# Action Typicality and Uniqueness Learning for Zero-Shot Video Anomaly Detection

## Abstract

Zero-Shot Video Anomaly Detection (ZS-VAD) is an urgent task in scenarios where the target video domain lacks training data due to various concerns, *e.g.*, data privacy. The skeleton-based approach is a promising way to achieve ZS-VAD as it eliminates domain disparities in both background and human appearance. However, existing methods only learn low-level skeleton representation and rely on the domain-specific normality boundary, which cannot generalize well to new scenes with different normal and abnormal behavior patterns. In this paper, we propose a novel skeleton-based zero-shot video anomaly detection framework, which captures both scene-generic typical anomalies and scene-adaptive unique anomalies. Firstly, we introduce a language-guided typicality modeling module that projects skeleton snippets into action semantic space and learns generalizable typical distributions of normal and abnormal behavior. Secondly, we propose a test-time context uniqueness analysis module to finely analyze the spatio-temporal differences between skeleton snippets and then derive scene-adaptive boundaries. Without using any training samples from the target domain, our method achieves state-of-the-art results on four large-scale VAD datasets: ShanghaiTech, UBnormal, NWPU, and UCF-Crime. The Code will be publicly available.

## 1 Introduction

Video Anomaly Detection (VAD) aims to temporally locate abnormal events, which has wide applications in the context of video surveillance and public safety (Liu et al., 2018; Sultani et al., 2018). Current mainstream paradigms include one-class (Wang et al., 2022; Liu et al., 2021; Hirschorn & Avidan, 2023) and weakly supervised methods (Sultani et al., 2018; Cho et al., 2023; Shi et al., 2023), which require samples from the target video domain for training. However, in surveillance scenarios involving privacy or newly installed monitoring devices, training samples from the target domain are often not available. Therefore, designing a Zero-Shot Video Anomaly Detection (ZS-VAD) method is necessary, as shown in Fig. 1 (a).

Despite the success of zero-shot anomaly detection in image domain (Zhou et al., 2024; Cao et al., 2024), only a few works (Aich et al., 2023; Guo et al., 2024) have ventured into the more complex video domain with underwhelming performance. Although recently (Zanella et al., 2024) proposes to leverage large visual language models to tackle the ZS-VAD task, it relies on multi-stage reasoning and the coordination of multiple large models, posing challenges for widespread deployment. Compared with them, we find skeleton-based method (Hirschorn & Avidan, 2023; Flaborea et al., 2023) is a promising way to achieve ZS-VAD as they are computation-friendly and can exclude the domain gap in human appearance and background. Existing methods use skeleton prediction, reconstruction, or coordinate-based normalizing flow to learn the normal skeleton distribution. However, they can only learn low-level spatio-temporal representation and rely on domain-specific normality boundaries, which cannot generalize well to new scenes with different normal and abnormal patterns. Within this line of work, ZS-VAD seems to become an ill-posed problem, because the new scene's decision boundary is unattainable, as illustrated in Fig. 1 (b). This limitation leads to a question: "*Can we design a model that can generalize to various target scenes with generalizable representation learning and prior injection?*"

We reflect on how human observers judge normal and abnormal behavior in a new scenario, as illustrated in Fig. 1 (c). Initially, we identify the types of actions performed by individuals in the

Figure 1: (a) The illustration of zero-shot video anomaly detection, where the model is tasked with localizing anomalous events in various scenes without any training samples from the target domain. (b) Previous methods do not involve action semantics and their learned normality boundaries in source data cannot generalize well to new scenes with different normal and anomalous behavior. (c) The illustration of how human observers or our model judge normal and abnormal in new scenarios.

video and consider whether they are normal or abnormal based on our experiential knowledge of normality and abnormality (*typicality*). For instance, a pedestrian walking would be considered normal, while a fight or scuffle would be deemed abnormal. Secondly, for atypical normal or abnormal scenarios, we integrate the behaviors of all individuals in the video to observe if any individual's behavior significantly differs from others, as anomalies are usually rare and unique (*uniqueness*).

Based on these complementary priors, we propose a novel skeleton-based zero-shot video anomaly detection framework, which captures both typical anomalies guided by language prior and unique anomalies in spatio-temporal contexts. Firstly, we introduce a language-guided typicality modeling module that projects skeleton snippets into a semantic space and learns typical distributions of normal and abnormal behavior. To obtain the semantic representations of skeleton snippets, we align skeleton snippets with CLIP's language embeddings using external skeleton action recognition datasets. Furthermore, by leveraging human experiential typicality labels obtained from language model, we train a feature normalization flow to learn domain-general boundaries between normal and abnormal behavior. Secondly, we propose a context uniqueness analysis module at test time to derive scene-adaptive boundaries. To gain a fine-grained understanding of the scene context, we construct two types of spatio-temporal context graphs: a cross-person graph and a self-inspection graph. The uniqueness scores are derived within these graphs, indicating whether the current skeleton is unique in the surroundings or if there is a sudden change in the motion state. Without using any training samples from the target domain, we achieved state-of-the-art results on four large-scale VAD datasets: ShanghaiTech (Liu et al., 2018), UBnormal (Acsintoae et al., 2022), NWPU (Cao et al., 2023), UCF-Crime (Sultani et al., 2018). Our contributions are as follows:

- We propose a new setting for skeleton-based zero-shot video anomaly detection and introduce a novel video anomaly detection framework that can generalize to various target scenes with action typicality and uniqueness learning.

- We propose a language-guided typicality modeling module that projects skeleton snippets into a generalizable semantic space and effectively learns the typical distribution of normal and abnormal behavior based on human experiential knowledge.

- We propose a test-time uniqueness analysis module to finely analyze the spatio-temporal differences between skeleton snippets and derive scene-adaptive boundaries between normal and abnormal behavior.

## 2 RELATED WORK

**Video anomaly detection.** Most previous video anomaly detection studies can be grouped into frame-based (Liu et al., 2018; 2021; Sultani et al., 2018), object-centric (Wang et al., 2022; Sun & Gong, 2023; Micorek et al., 2024), and skeleton-based methods (Markovitz et al., 2020; Morais

et al., 2019; Flaborea et al., 2023; Hirschorn & Avidan, 2023). In this work, we focus on the skeleton-based methods, which detect anomalies in human activity based on preprocessed skeleton/pose snippets. Morais et al. (Morais et al., 2019) propose an anomaly detection method that uses RNN to learn the representation of pose snippets, with prediction errors serving as anomaly scores. GEPC (Markovitz et al., 2020) utilizes autoencoders to learn pose graph embeddings, generates soft assignments through clustering, and uses Dirichlet process mixture to determine anomaly scores. To model normal diversity, MoCoDAD (Flaborea et al., 2023) leverages diffusion probabilistic models to generate multimodal future human poses. STG-NF (Hirschorn & Avidan, 2023) proposes a simple yet effective method by establishing normalized flow from normal pose snippets to obtain normal boundaries. However, these methods rely on training with normal data from the target domain, while overlooking the semantic understanding of human behavior, which makes it difficult to ensure performance when the data is unavailable.

**Zero-shot anomaly detection.** Thanks to the development of vision-language models, zero-shot anomaly detection has received a lot of attention (Jeong et al., 2023; Li et al., 2024; Gu et al., 2024; Aota et al., 2023; Liu et al., 2024; Chen et al., 2023; Zhou et al., 2024), especially in the field of image anomaly detection (Miyai et al., 2024). The pioneering work is WinCLIP (Jeong et al., 2023), which utilizes CLIP (Radford et al., 2021)'s image-text matching capability to distinguish between unseen normal and abnormal anomalies. Building on that, AnomalyCLIP (Zhou et al., 2024) proposes to learn object-agnostic text prompts that capture generic normal and abnormal patterns in an image. AdaCLIP (Cao et al., 2024) introduces two types of learnable prompts to enhance CLIP's generalization ability for anomaly detection data. Despite the success in the image domain, only a few works (Aich et al., 2023; Guo et al., 2024) have ventured into zero-shot video anomaly detection with underwhelming performance. Although recently (Zanella et al., 2024) proposes to leverage large visual language models for zero-shot video anomaly detection, it requires multi-stage reasoning and the collaboration of multiple large models, making it less user-friendly. We aim to develop a lightweight, user-friendly, and easily deployable zero-shot anomaly detector starting from skeleton data. Our work shares some similarities with a recent study (Sato et al., 2023). However, we emphasize that our approach differs significantly from (Sato et al., 2023) in the following ways: **1) Different tasks:** (Sato et al., 2023) addresses abnormal action recognition, involving no more than two individuals in a short video, without the need for temporally localizing abnormal events. **2) Novel perspective:** We combine the action typicality and uniqueness priors to address zero-shot anomaly detection challenges in video surveillance scenarios.

## 3 ACTION TYPICALITY AND UNIQUENESS LEARNING

### 3.1 OVERVIEW

The objective of ZS-VAD is to train one model that can generalize to diverse target domains. Formally, let $\mathcal{V}^{train}$ be a training set from source video domain and $\{\mathcal{W}_1^{test}, \mathcal{W}_2^{test}, ..., \mathcal{W}_N^{test}\}$ be multiple test sets from target video domain. The test videos are annotated at the frame level with labels $l_i \in \{0, 1\}$, and the VAD model is required to predict each frame's anomaly score. In this work, we focus on the skeleton-based paradigm, as it is computation-friendly and can benefits ZS-VAD by reducing the domain gap in both background and appearance.

Fig. 2 overviews our proposed approach. Our model tackles the ZS-VAD problem from the perspective of action typicality and uniqueness learning. First, the Language-Guided Typicality Modeling module projects skeleton snippets into a semantic space and effectively learns the typical distribution of normal and abnormal behavior based on experiential knowledge from the language model. Second, the Test-Time Uniqueness Analysis module finely analyzes the spatio-temporal differences between skeleton snippets and derives scene-adaptive boundaries. During inference on unseen VAD datasets, our model integrates typicality scores and uniqueness scores of human behavior to provide a holistic understanding of anomalies.

### 3.2 LANGUAGE-GUIDED TYPICAL MODELING

This module aims to obtain a generalizable semantic understanding of human behavior in videos by learning discriminative representation of skeleton snippets. Furthermore, it learns the scene-generic distributions of typical normal and abnormal behavior by leveraging the prior knowledge

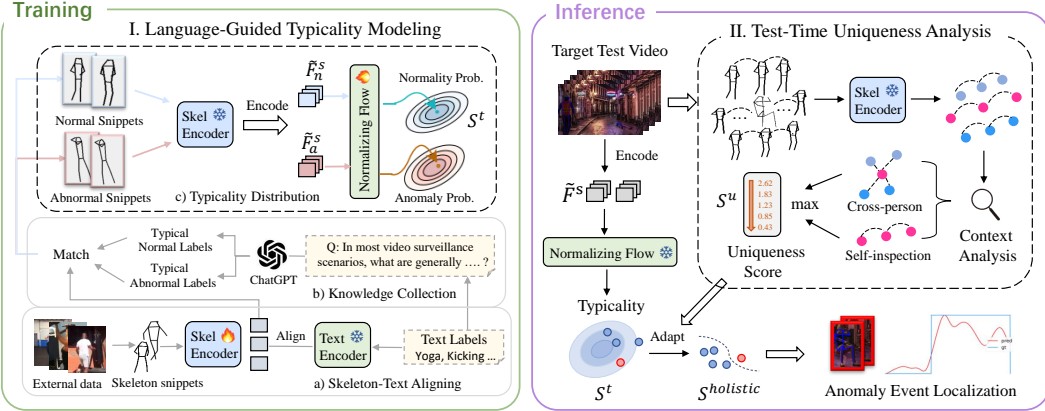

Figure 2: Overview of our approach for skeleton-based zero-shot video anomaly detection. **I.** Language-guided typicality modeling in the training phase. It projects skeleton snippets into the action semantic space, collects typicality knowledge from language model, and then effectively learns the typical distribution of normal and abnormal behavior. (Only the black dashed boxes are used during inference.) **II.** Test-time uniqueness analysis in the inference phase. It finely analyzes the spatio-temporal differences between skeleton snippets and derives scene-adaptive boundaries between normal and abnormal behavior.

acquired from language models. Specifically, this module consists of skeleton-text alignment, typicality knowledge selection, and typicality distribution learning.

**Skeleton-text alignment.** For achieving a generalizable semantic understanding of human behavior, we first propose to align the skeleton snippets with the corresponding semantic labels. For such skeleton-text pairs, we utilize external action recognition datasets (e.g., Kinect (Carreira & Zisserman, 2017) as the training set instead of specific VAD datasets (e.g., ShanghaiTech). The raw skeleton data of an action video is typically formally represented as $\mathbf{X}_i \in \mathbb{R}^{C \times J \times L \times M}$, where $C$ is the coordinate number, $J$ is the joint number, $L$ is the sequence length, and $M$ is the pre-defined maximum number of persons, respectively. In addition, each video is annotated with a text label $g_i$ representing the action class, which can be also transformed into a one-hot class vector $\mathbf{y}_i$.

Compared to action recognition tasks that only predict video-level categories, the VAD task is more fine-grained, focusing on predicting frame-level anomaly scores. Therefore, we decompose the original sequences into multiple short skeleton snippets $\mathbf{A}_i \in \mathbb{R}^{C \times J \times T}$ using a sliding window, and discard snippets that are predominantly composed of zeros. For the snippets from the same action video, they share the same labels and undergo a normalization operation to make different snippets independent. Inspired by the recent multimodel alignment works (Wang et al., 2021; Xiang et al., 2023), we then perform a skeleton-text alignment pretraining procedure to learn the discriminative representation. The procedure is built with a skeleton encoder $E^s$ and a text encoder $E^t$, for generating skeleton features $\mathbf{F}^s$ and text features $\mathbf{F}^t$, respectively. Additionally, the skeleton encoder also predicts a probability vector $\hat{\mathbf{y}}_i$ using a fully-connected layer. The training loss consists of a KL divergence loss and a cross-entropy classification loss, which are presented as follows:

$$\mathcal{L}_s = \frac{1}{2N} \sum_{i=1}^{N} [\text{KL}(p^{s2t}(\mathbf{F}_i^s), y_i^{s2t}) + \text{KL}(p^{t2s}(\mathbf{F}_i^t), y_i^{t2s})] + \mathcal{L}_{cls}(\hat{\mathbf{y}}, \mathbf{y}), \quad (1)$$

where $N$ is the sample count. More details can be found in the appendix.

**Typicality knowledge selection.** In most video surveillance scenarios, some behaviors are generally considered normal or abnormal, which constitute a scene-generic set. Therefore, training a typicality-aware capability is one of the promising ways to achieve ZS-VAD. Based on the pretrained skeleton-text representation, we aim to use a Large Language Model (LLM) as our knowledge engine to collect typical normal and abnormal data from the massive skeleton snippets. In detail, we give the large model a prompt $\mathcal{P}$: "*In most video surveillance scenarios, what are generally considered as normal actions and abnormal actions among these actions: $<\mathcal{T}>$...*", where $\mathcal{T}$ refers to the set of all action class labels in the action recognition dataset. The large language model

will respond with a list of typical normal action classes $\mathcal{T}^n$ and a list of typical abnormal action classes $\mathcal{T}^a$, which can be formalized as:

$$\mathcal{T}^n, \mathcal{T}^a = \mathrm{O}_{LLM}(\mathcal{P}, \mathcal{T}), \tag{2}$$

where $\mathcal{T}^n$ and $\mathcal{T}^a$ are the subsets of $\mathcal{T}$, and $\mathcal{O}_{LLM}$ denotes the offline LLM. Note that the LLM is only needed to be used once during training for auxiliary data selection, while inference is not. Therefore, we use the powerful LLM, ChatGPT (OpenAI, 2022), as our knowledge engine, for initial typical label generation.

After knowing the action categories of typicality, we first collect the data of these selected categories and then proceed to select the high-quality snippets from them. This is because 1) Some snippets contain noise, such as errors in pose detection and tracking. 2) In an abnormal action sequence, not all the snippets are abnormal. Therefore, we use the skeleton-text similarity score to filter the skeleton snippets, which is formulated as:

$$\mathcal{M}^x = \{\arg \mathop{\mathrm{top}}\limits_{i} \beta^x \left( \mathrm{sim}\left( \mathbf{F}_i^s, \mathbf{F}_j^t \right) \right) \mid g_i \in \mathcal{T}^x, g_j \in \mathcal{T}^x\}, \tag{3}$$

where $\mathcal{M}^x$ refers to the selected snippets index and $\beta$ denotes the selection ratio. The superscript $x$ represents $n$ or $a$, indicating normal and abnormal, respectively. Using the index $\mathcal{M}^x$, we obtain the corresponding skeleton data $\tilde{\mathbf{A}}^n$ and $\tilde{\mathbf{A}}^a$, as well as skeleton features $\tilde{\mathbf{F}}_n^s$ and $\tilde{\mathbf{F}}_a^s$.

**Typicality distribution learning.** After obtaining the data, we move on to model the feature distribution of typical behavior. Normalizing Flow (NF) (Kingma & Dhariwal, 2018) provides a robust framework for modeling feature distributions, transforming this distribution through a series of invertible and differentiable operations. Consider a random variable $\mathbf{X} \in \mathbb{R}^D$ with target distribution $p_X(x)$, and a random variable Z follows a spherical multivariate Gaussian distribution. Introducing a bijective map $f : X \leftrightarrow Z$, composed of a sequence of transformations: $f_1 \circ f_2 \circ ... \circ f_K$. According to the variable substitution formula, the log-likelihood of X can be expressed as:

$$\log p_X(x) = \log p_Z(f(x)) + \sum\nolimits_{i=1}^{K} \log \left| \det(\frac{df_i}{df_{i-1}}) \right|. \tag{4}$$

The bijective maps for the normal features and abnormal features are $f : X_n \leftrightarrow Z_n$ and $f : X_a \leftrightarrow Z_a$, respectively. Here, the log-likelihood of $Z_n$ and $Z_a$ are as follows:

$$\log p_{Z_n}(z) = \mathrm{Con} - \frac{1}{2}(z - \mu_n)^2, \quad \log p_{Z_a}(z) = \mathrm{Con} - \frac{1}{2}(z - \mu_a)^2, \tag{5}$$

where Con is a constant, and $u_n$ and $u_z$ are the centers of the Gaussian distributions ($|u_n - u_z| \gg 0$), respectively. During training, the normalizing flow is optimized to increase the log-likelihood of the skeleton features $\mathbf{F}^s$ with the following loss:

$$\mathcal{L}_n = -\log p_{X_n}(\mathbf{F}_n^s) - \log p_{X_a}(\mathbf{F}_a^s), \tag{6}$$

During inference, the testing skeleton snippet $\mathbf{F}_i^s$ will be sent to the trained normalizing flow, outputting the typicality anomaly score as follows:

$$\mathbf{S}_i^t = -\log p_{X_n}(\mathbf{F}_i^s), \tag{7}$$

where the normal skeletons will exhibit low $\mathbf{S}_i^t$, while the anomalies will exhibit higher $\mathbf{S}_i^t$. Our approach differs significantly from STG-NF (Hirschorn & Avidan, 2023). It takes low-level joint coordinates as inputs and only learns implicit spatio-temporal features, which struggle to generalize to new datasets without the normality reference of training data from the target dataset. Differently, we use action semantics as a generalizable representation for NF input, and leverage experiential typicality labels to learn domain-general boundaries between normal and abnormal behavior.

### 3.3 TEST-TIME UNIQUENESS ANALYSIS

The goal of this component is to serve as a complementary perspective of typicality, deriving scene-adaptive boundaries by considering the context of the target scene. To this end, we propose a context uniqueness analysis module during the inference of the unseen VAD dataset.

Unlike action recognition datasets, surveillance videos often contain a much longer temporal span, involve a larger number of people, and exhibit more diverse behavioral patterns. For such a video, we

first extract $H$ skeleton sequences $\{\mathbf{X}_1, ..., \mathbf{X}_H\}$, where each sequence comprises $L_i$-frame poses, represented as $\mathbf{X}_i = \{\mathbf{P}_1, ..., \mathbf{P}_{L_i}\}$. Here, $\mathbf{P}_t \in \mathbb{R}^{J \times 2}$ comprises $J$ keypoints, each defined by a pair of coordinate values. Targeted at frame-level anomaly scoring, the sequences are then segmented into shorter skeleton snippets $\mathbf{A}_i \in \mathbb{R}^{C \times J \times T}$ as described in Section 3.2.

**Spatio-temporal context.** To gain a fine-grained context understanding of the scene, we construct two types of spatio-temporal context graphs: a cross-person graph $\mathcal{G}^c$ and a self-inspection graph $\mathcal{G}^s$. The first graph is constructed by retrieving the feature nearest neighbors among the surrounding skeleton snippets, while the second one is constructed by retrieving the feature nearest neighbors from different time segments of the current person. In this way, we can filter out some unrelated activities and focus solely on behaviors related to the current individual. Given a skeleton snippet $\mathbf{A}_i$ with feature $\mathbf{F}_i^s$, the cross-person graph is defined as $\mathcal{G}_i^c = \{\mathcal{V}_i^c, \mathcal{E}_i^c\}$, where $\mathcal{V}_i^c = \{\mathbf{A}_i, \mathcal{N}_c(\mathbf{A}_i)\}$ denotes the node set and $\mathcal{E}_c^i = \{(i, j) \mid j \in \mathcal{N}_c\}$ denotes the edge set. Besides, $\mathbf{A}_i$ is associated with a person index $p_i$ and timestamp $t_i$. The neighborhood $\mathcal{N}_c$ is formulated as:

$$\mathcal{N}_c = \left\{ \mathbf{A}_j \mid d\left(\mathbf{F}_i^s, \mathbf{F}_j^s\right) \leq \text{topk}\left(d\left(\mathbf{F}_i^s, \mathbf{F}_j^s\right)\right), p_j \neq p_i \right\}, \tag{8}$$

where $d(\cdot)$ represents the Euclidean distance, and topk refers to the k-th smallest value for the self-person graph, it is defined as $\mathcal{G}_i^s = \{\mathcal{V}_i^s, \mathcal{E}_i^s\}$, where $\mathcal{V}_i^s = \{\mathbf{A}_i, \mathcal{N}_s(\mathbf{A}_i)\}$ denotes the node set and $\mathcal{E}_s^i = \{(i, j) \mid j \in \mathcal{N}_s\}$ denotes the edge set. Then, the neighborhood $\mathcal{N}_s$ is formulated as:

$$\mathcal{N}_s = \{\mathbf{A}_j \mid d\left(\mathbf{F}_i^s, \mathbf{F}_j^s\right) \leq \text{topk}\left(d\left(\mathbf{F}_i^s, \mathbf{F}_j^s\right)\right), p_j = p_i, |t_i - t_j| > \alpha \cdot T\}, \tag{9}$$

where $\alpha$ is a threshold that masks out a period of time before and after the current time window, as the individual's state tends to remain stable during adjacent periods.

**Uniqueness scores.** Since abnormal activities are rare, anomalies in real-world surveillance videos often differ from other activities at most times, *i.e.*, uniqueness. Based on the pre-trained discriminative skeleton features, uniqueness can be represented as the feature distances between a query node and other nodes in the built graph. Specifically, the uniqueness score $\mathbf{S}^u$ for individual $i$ is obtained by taking the larger one of the cross-person and self-inspection distances, formulated as follows:

$$\mathbf{S}_i^u = \max \left\{ \sum_{j \in \mathcal{N}_c(\mathbf{A}_i)} d\left(\mathbf{F}_i^s, \mathbf{F}_j^s\right), \sum_{j \in \mathcal{N}_s(\mathbf{A}_i)} d\left(\mathbf{F}_i^s, \mathbf{F}_j^s\right) \right\}. \tag{10}$$

**Holistic anomaly scoring.** By integrating the complementary typicality $\mathbf{S}_i^t$ scores and the uniqueness scores $\mathbf{S}_i^u$, our model can capture both typical anomalies in language prior and unique anomalies in spatio-temporal contexts. This helps gain a comprehensive understanding of anomalies in new scenes, where the holistic anomaly score of individual $i$ is defined as:

$$\mathbf{S}_i = \frac{\mathbf{S}_i^t - \mathbf{S}_{mean}^t}{\mathbf{S}_{std}^t} + \frac{\mathbf{S}_i^u - \mathbf{S}_{mean}^u}{\mathbf{S}_{std}^u}. \tag{11}$$

In the end, the frame-level anomaly score is obtained by taking the highest score among all individuals within that frame. If any individual is considered anomalous, the entire frame is classified as anomalous. Following (Hirschorn & Avidan, 2023), when no individuals are detected in the frame, it is considered normal, with the score being the minimum among all anomaly scores for that video.

# 4 EXPERIMENTS

## 4.1 DATASET AND IMPLEMENTATION DETAILS

**Dataset.** The training of our model is conducted on the Kinect-400 dataset (Carreira & Zisserman, 2017), while the ZS-VAD capability of our model is evaluated on four large-scale VAD datasets: ShanghaiTech (Liu et al., 2018), UBnormal (Acsintoae et al., 2022), NWPU (Cao et al., 2023) and UCF-Crime (Sultani et al., 2018). Note that we only use the test set of these four VAD dataset. **(1)** Kinect-400: It is not intended for VAD tasks but for action recognition, which is gathered from YouTube videos covering 400 action classes. We utilize the preprocessed skeleton data obtained from ST-GCN (Yan et al., 2018) for training. **(2)** VAD datasets for evaluation: *ShanghaiTech* contains 107 test videos from 13 different scenes. *UBnormal* includes 211 test videos from 29 different scenes. *NWPU* is a newly published dataset containing 242 testing videos from 43 scenes. *UCF-Crime* includes 290 testing videos from more than 50 scenes. Appendix contains more details.

Table 1: Zero-shot video anomaly detection performance on the four large-scale datasets, ShanghaiTech (SHT), UBnormal (UB), NWPU, and UCF-Crime (UCFC), where the subscript denotes the number of scenes of the dataset. The evaluation metric is frame-level AUC (%). "LVLM (infer.)" indicates using Large-Vision-Language-Models during inference.

| paradigm | Method | Venue | LVLM (infer.) | Training VAD | Testing | | | |
|---|---|---|---|---|---|---|---|---|
| | | | | | $SHT_{13}$ | $UB_{29}$ | $NWPU_{43}$ | $UCFC_{>50}$ |
| LVLM | Imagebind | CVPR'23 | ✔ | ✗ | - | - | - | 55.8 |
| | LAVAD | CVPR'24 | ✔ | ✗ | - | - | - | 80.3 |
| frame/object | HF2-VAD | ICCV'21 | ✗ | SHT | - | 59.5 | 58.3 | 52.9 |
| | Jigsaw-VAD | ECCV'22 | ✗ | SHT | - | 58.6 | 61.1 | 53.3 |
| skeleton | MocoDAD | ICCV'23 | ✗ | SHT | - | 67.6 | 56.4 | 51.8 |
| | MocoDAD | ICCV'23 | ✗ | UB | 76.0 | - | 56.6 | 52.0 |
| | STG-NF | ICCV'23 | ✗ | SHT | - | 68.8 | 57.6 | 51.6 |
| | STG-NF | ICCV'23 | ✗ | UB | 83.0 | - | 57.9 | 51.9 |
| | **Ours** | - | ✗ | ✗ | **84.1** | **74.5** | **62.1** | **62.7** |

**Implementation Details.** For a fair comparison, we directly use the skeleton data of ShanghaiTech and UBnormal from STG-NF (Hirschorn & Avidan, 2023). For NWPU and UCF-Crime, as they do not have open-source skeleton data, we resort to utilizing AlphaPose (Fang et al., 2022) for data extraction. We use a segment window $T = 16$ and a stride of 1 to divide each sequence into snippets. Specifically, we use a stride of 16 for UCF-Crime because its videos are too long. The batch size is set to 1024, and we use Adam as the optimizer with a

Table 2: Our zero-shot performance vs. SOTA full-shot performance.

| Setting | Method | SHT | UB |
|---|---|---|---|
| full-shot | Jigsaw-VAD | 84.3 | 60.9 |
| | MocoDAD | 77.6 | 68.3 |
| | STG-NF | **85.9** | 71.8 |
| zero-shot | **Ours** | 84.1 | **74.5** |

learning rate of 0.0005. Additionally, the hyperparameters $\beta^n$, $\beta^a$, $k$, and $\alpha$ are set to 90%, 10%, 16, and 4, respectively. For the evaluation metrics, we follow common practice (Liu et al., 2018; Sultani et al., 2018; Hirschorn & Avidan, 2023) by using the micro-average frame-level AUC as the evaluation metric, which involves concatenating all frames and calculating the score. More details can be found in appendix.

## 4.2 ZS-VAD PERFORMANCE

We conduct a comprehensive comparison of the performance of ZS-VAD, comparing the frame-based/object-based (Liu et al., 2021; Wang et al., 2022), skeleton-based (Flaborea et al., 2023; Hirschorn & Avidan, 2023), and LVLM-based methods (Girdhar et al., 2023; Zanella et al., 2024).

**Comparison with frame/object-based methods.** We use their open-source checkpoints trained on the ShanghaiTech to evaluate the zero-shot performance on the remaining three VAD datasets. As shown in Tab. 1, their generalization capabilities on new scene datasets are relatively poor due to the influence of human appearance and background variations.

**Comparison with skeleton-based methods.** We use their open-source checkpoints trained on the ShanghaiTech or UBnormal to evaluate on the remaining three VAD datasets. The performance of prevalent skeleton-based methods is still underwhelming due to a lack of understanding of complex normal and abnormal behaviors without target training data. Compared with our baseline STG-NF, our proposed method improves the frame-level AUC-ROC by 1.1% on ShanghaiTech, 5.7% on UBnormal, 4.2% on NWPU, and 10.8% on UCF-Crime. We also compare their performance in the full-shot setting, where target domain data is used for training. Table 2 shows that our zero-shot approach can achieve comparable or even superior results to SOTA full-shot performance.

**Comparison with LVLM-based methods.** While LAVAD (Zanella et al., 2024) recently proposes leveraging large visual language models for ZS-VAD, it relies on multi-stage reasoning and the coordination of multiple large models with over **13 billion (B)** parameters, posing challenges for widespread deployment. In contrast, we develop a lightweight zero-shot anomaly detector with a mere **5.0 million (M)** parameters, just one in two thousand of LAVAD's parameters.

Table 3: Ablation experiments of the typicality modeling module.

| Experiments | SHT | UB | NWPU | UCFC |
|---|---|---|---|---|
| (a) ours w/o aligning | 83.2 | 69.4 | 60.4 | 59.5 |
| (b) ours w/o selection | - | 64.1 | 61.9 | 56.7 |
| (c) ours w/o NF | 83.4 | 72.2 | **62.6** | 60.5 |
| (d) prompt score | 81.3 | 64.4 | 61.5 | 61.0 |
| (e) ours | **84.1** | **74.5** | 62.1 | **62.7** |

Table 4: Ablation study of the uniqueness analysis module and holistic anomaly scoring.

| Typ. | Cross | Self | SHT | UB | NWPU | UCFC |
|---|---|---|---|---|---|---|
| ✓ | | | 81.9 | 73.2 | 62.1 | 59.6 |
| | ✓ | | 81.9 | 62.9 | 60.7 | 59.9 |
| | | ✓ | 67.8 | 60.1 | 61.0 | 62.6 |
| | ✓ | ✓ | 82.0 | 64.5 | 61.7 | 61.0 |
| ✓ | ✓ | ✓ | **84.1** | **74.5** | 62.1 | **62.7** |

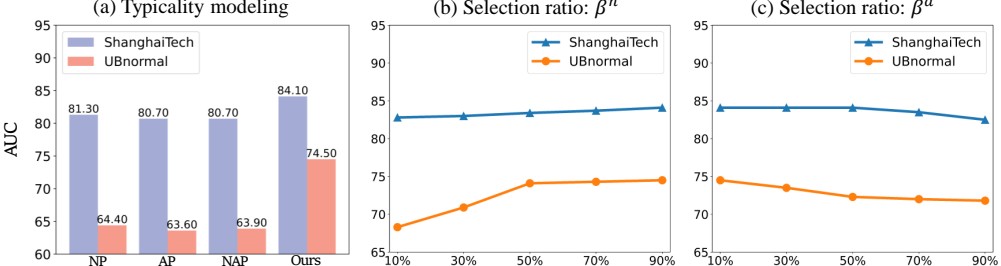

Figure 3: (a) Comparison between our typicality modeling module and the prompt-based schemes. (b)-(c) Comparison of different selection ratios in the typicality knowledge selection step.

## 4.3 ABLATION STUDY

**Ablation of typicality module.** We conduct ablation experiments on the typicality modeling module with the following settings: (a) removing the aligning stage, training with raw skeleton data and the typicality labels; (b) removing the collection phase, training with VAD source data (SHT); (c) removing the normalizing flow and calculating typicality scores using k-nearest neighbors distance techniques. As shown in Table 3, the model shows poor performance without the aligning stage, as it fails to learn generalizable and discriminative semantic representations. Moreover, performance deteriorates without the selection of typicality knowledge, as the model can only learn a limited normality boundary from VAD source data. Furthermore, without the normalizing flow, the model also loses flexibility in modeling the distribution of typical behaviors.

**Comparison with prompt-based methods.** Since prompt-based techniques have been popular in other zero-shot tasks (Sato et al., 2023; Jeong et al., 2023), we conduct experiments to compare our typicality module with theirs. To this end, we design typical normal prompts (NP), typical abnormal prompts (AP), and the ensemble (NAP), then using the skeleton-prompt similarity as the anomaly score. More details can be found in the appendix. As shown in Table 3 (d) and Fig. 3 (a), the results are suboptimal. Unlike various forms of text seen in CLIP image-text alignment, the current skeleton-text alignment scheme has only encountered text of action class names, thus the alignment capability for prompt text is relatively weak. Our method, on the other hand, leverages LLM to collect typicality labels, avoiding directly using the skeleton-prompt similarity as anomaly scores.

**Ablation of uniqueness module.** We ablate the uniqueness scores and the holistic scores in this part. As demonstrated in Table 4, when only using the cross-person distance, the model can identify contextual anomalies with acceptable performance. When combined with the self-inspection score, the model can spot changes in motion states, aiding in detecting a wider range of anomalies. The reason for the suboptimal performance of uniqueness score on UBnormal is that UBnormal is a synthetic dataset where some videos contain only one person with relatively short movement durations, which does not align well with real surveillance video scenarios. By integrating both the typicality and uniqueness modules, our approach can achieve optimal performance.

**Hyper-parameter sensitivity.** We ablate the hyper-parameters in the typicality knowledge selection step. As shown in Fig. 3 (b)-(c), the optimal value of $\beta^n$ is 90%, and a smaller $\beta^a$ can enhance performance by filtering out noisy data and normal snippets within the anomalous sequences.

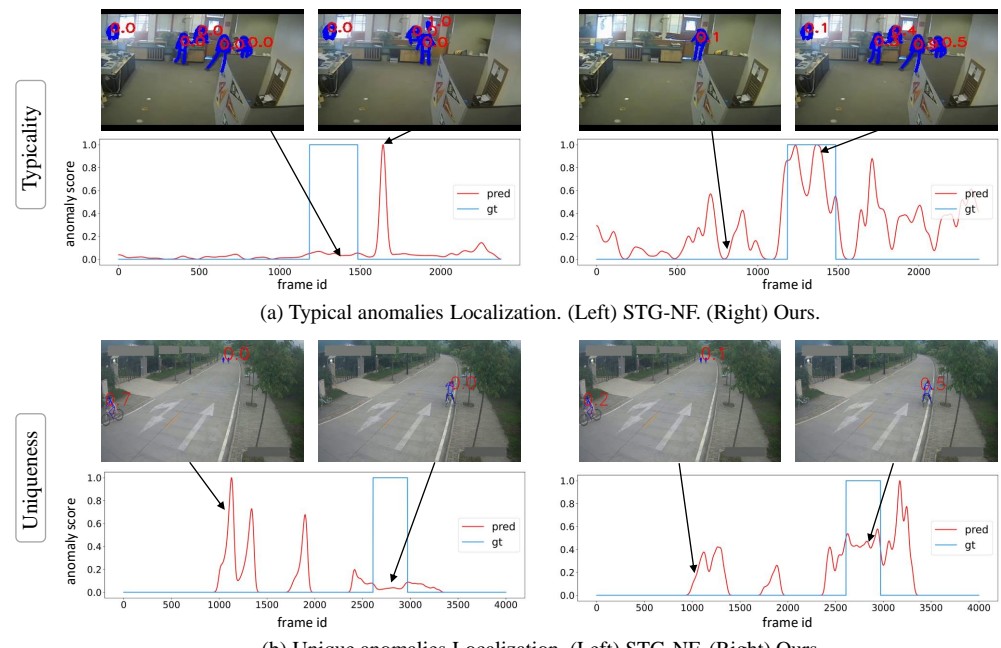

(a) Typical anomalies Localization. (Left) STG-NF. (Right) Ours.

(b) Unique anomalies Localization. (Left) STG-NF. (Right) Ours.

Figure 4: Example results of our method that succeeds in capturing typical and unique anomalies. For the "arresting" and "photographing at restricted areas" anomalies, STG-NF (Hirschorn & Avidan, 2023) fails to detect them, while ours performs well through action typicality and analysis modeling. Each individual (blue skeleton) has a predicted anomaly score (red font), where the frame-level score (red line) is defined as the maximum among all individuals in that frame. **More visualization results can be found in the appendix.**

## 4.4 VISUALIZATION RESULTS

Fig. 4 (a) presents the qualitative localization results of typical anomalies. STG-NF fails to detect the "arresting" anomaly because of its insufficient discriminative capability for action semantics. Disturbed by around 1600 frames of joint noise, it erroneously positioned the anomaly. In contrast, our model can map the skeleton snippets to a space with generalizable semantics and identify the anomaly of the "arrest" action based on our training experience boundary. In surveillance scenarios lacking training samples, our model can still be effectively utilized to detect certain typical abnormal behaviors that pose a potential for harm.

Fig. 4 (b) shows the qualitative localization results of unique anomalies. Existing skeleton-based methods rely on the source normal data for training. When the source domain does not include some novel behavior that appears in the target domain, these behavior will be classified as anomalies. Consequently, STG-NF erroneously localizes the anomaly during time periods when "riding" is present. In contrast, our model can analyze the spatio-temporal differences and establish scene-adaptive decision boundaries. Since "riding a bicycle" occurs multiple times in the video, its uniqueness score is relatively low. On the other hand, "photographing at restricted areas" exhibits significant differences from the surrounding people's behavior and appears as a sudden change in the person's movement trajectory, resulting in a corresponding increase in its anomaly score.

## 4.5 CONCLUSION

In this paper, we introduce a novel ZS-VAD framework that can generalize to various target scenes with generalizable representation learning and prior injection. First, we propose a language-guided typicality modeling module that effectively learns the typical distribution of normal and abnormal behavior. Secondly, we propose a test-time uniqueness analysis module to derive scene-adaptive boundaries. Comprehensive experiments demonstrate the effectiveness of our model.

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

# A APPENDIX

## A.1 MORE IMPLEMENTATION DETAILS

**Backbone.** We use multi-scale CTR-GCN (Chen et al., 2021) (2.1M) as the skeleton encoder and use the text encoder of 'ViT-B/32' CLIP (Radford et al., 2021) (63.4M). During inference, the text encoder is removed, and the trained skeleton encoder is retained to extract skeleton features.

**Alignment Loss.** The training loss $\mathcal{L}_s$ employed in skeleton-text aligning (Sec. 3.2) is inspired by GAP (Xiang et al., 2023), which consists of a KL divergence loss and a cross-entropy classification loss. As there could be more than one positive matching and actions of different categories forming negative pairs, GAP uses KL divergence as the alignment loss. In the KL divergence loss, $y^{s2t}$ and $y^{t2s}$ denote the ground-truth matching labels. Based on the loss function in CLIP (Radford et al., 2021), the networks are optimized by contrasting skeleton-text pairs in two directions within the batch:

$$p^{s2t}(\mathbf{F}_i^s) = \frac{\exp(\mathrm{sim}(\mathbf{F}_i^s, \mathbf{F}_i^t)/\tau)}{\sum_{j=1}^{B} \exp(\mathrm{sim}(\mathbf{F}_i^s, \mathbf{F}_j^t)/\tau)}, \quad p^{s2t}(\mathbf{F}_i^t) = \frac{\exp(\mathrm{sim}(\mathbf{F}_i^t, \mathbf{F}_i^s)/\tau)}{\sum_{j=1}^{B} \exp(\mathrm{sim}(\mathbf{F}_i^t, \mathbf{F}_j^s)/\tau)}, \quad (12)$$

where $B$ is the batch size, and $\mathrm{sim}(\cdot)$ denotes the cosine similarity.

**Prompt for ChatGPT.** During the typicality knowledge selection (Sec. 3.2), we provide ChatGPT with the prompt: "*In most video surveillance scenarios, what are generally considered as normal actions and abnormal actions among these actions (Please identify the 20 most typical normal actions and 20 most typical abnormal actions, ranked in order of decreasing typicality). The action list is $<\mathcal{T}>$*", where $\mathcal{T}$ refers to the set of all action class labels in the action recognition dataset.

**Typicality prompt design.** To compare our typicality learning module with prompt-based methods (Sato et al., 2023; Jeong et al., 2023) (Sec. 4.3), we design a normal prompt $P^n$ list: ["usual", "normal", "daily", "stable", "safe"], and a prompt $\mathcal{P}^a$ list: ["danger", "violence", "suddenness", "unusual", "instability"]. The prompt $\mathcal{P}^n$ is encoded into normal prompt features $\mathbf{F}_n^p$, and $\mathcal{P}^a$ is encoded into abnormal prompt features $\mathbf{F}_a^p$. Then, the anomaly scores derived from $\mathcal{P}^n$ and $\mathcal{P}^a$ are defined as follows:

$$\mathbf{S}_i^n = -\mathrm{sim}\left(\mathbf{F}_n^p, \mathbf{F}_i^s\right), \quad \mathbf{S}_i^a = \mathrm{sim}\left(\mathbf{F}_a^p, \mathbf{F}_i^s\right). \quad (13)$$

For the ensemble of the two types of prompt (NAP), the anomaly score $\mathbf{S}^e$ is obtained as follows:

$$\mathbf{S}_i^e = \frac{\exp\left(\mathrm{sim}(\mathbf{F}_a^p, \mathbf{F}_i^s)/0.07\right)}{\exp\left((\mathrm{sim}(\mathbf{F}_a^p, \mathbf{F}_i^s)/0.07) + \exp((\mathrm{sim}(\mathbf{F}_n^p, \mathbf{F}_i^s)/0.07)\right)}. \quad (14)$$

We then compare our typicality module with the three types of prompt schemes as in Sec. 4.3.

## A.2 DATASET

The ZS-VAD capability of our model is evaluated on four large-scale VAD datasets: ShanghaiTech (Liu et al., 2018), UBnormal (Acsintoae et al., 2022), NWPU (Cao et al., 2023) and UCF-Crime (Sultani et al., 2018). Some early VAD benchmarks (Lu et al., 2013; Li et al., 2013) consist of single scenes staged and captured at one location, whereas the four datasets we evaluated are more extensive, encompassing a wider variety of scenes. Consequently, these four

Table 5: The details of the zero-shot benchmarks.

| | Year | Resolution | Test Video Num. | Scenes Num. |
|---|---|---|---|---|
| ShanghaiTech | 2018 | 480×856 | 107 | 13 |
| UBnormal | 2022 | 720×1280 | 211 | 29 |
| NWPU | 2023 | multiple | 242 | 43 |
| UCF-Crime | 2018 | 240×320 | 290 | >50 |

datasets are better suited for testing the model's zero-shot capabilities and assessing its cross-scenario performance. The details are summarized in Table 5 and the following descriptions.

**ShanghaiTech.** It is a widely-used benchmark for one-class video anomaly detection, which consists of 330 training videos and 107 test videos from 13 different scenes at 480×856 pixel resolution.

**UBnormal.** It is a synthetic dataset with virtual objects and real-world environments. It consists of 186 training videos and 211 test videos from 29 different scenes at 720×1280 pixel resolution.

**NWPU.** It is a newly published dataset that contains some scene-dependent anomaly types. It comprises 305 training videos and 242 testing videos from 43 scenes with diverse resolutions.

**UCF-Crime.** It is a large-scale dataset with 1900 long untrimmed surveillance videos. The 290 testing videos at a resolution of 240×320 pixels are used for our evaluation. It is primarily used in weakly supervised settings (Sultani et al., 2018; Cho et al., 2023), where the training set contains both normal and abnormal videos. Following the ZS-VAD setting, we do not use any training samples at all, which significantly increases the anomaly detection difficulty on this dataset.

## A.3 HYPER-PARAMETER ABLATION

We ablate the nearest neighborhood (NN) number $k$ and the mask threshold $\alpha$ in the uniqueness analysis module. As shown in Fig. 5, our method is robust for these two hyper-parameters. Choosing an appropriate $k$ can filter out some unrelated activities and focus solely on behaviors related to the current skeleton snippets. Taking the average of the $k$ neighbors helps suppress noise, which also makes our model insensitive to $\alpha$.

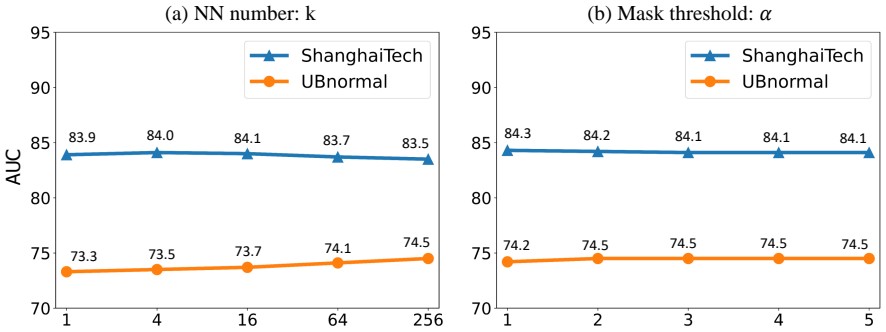

Figure 5: Comparison of different NN numbers and different mask thresholds.

## A.4 LIMITATION

**Skeleton detection and tracking error.** Obtaining skeleton data requires preprocessing by a skeleton/pose detector and tracker, which will introduce errors in some scenarios. As shown in Fig. 6 (left), the detector extracts an unstable skeleton sequence from a video of a stationary motorcycle, resulting in a high abnormal score for what should have been a normal segment. This is a common issue with skeleton-based methods, which can be addressed in the future by using more robust detectors and trackers.

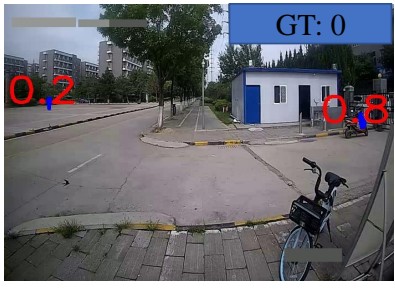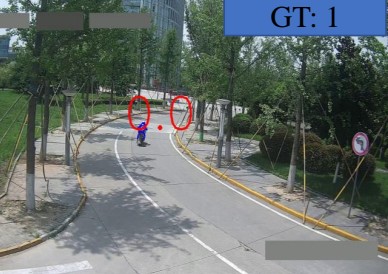

Figure 6: The failure cases of our model. (Left) The case's failure is the skeleton detection error. (Right) The case's failure is due to no reference individuals in the video.

**Scenarios with only one person.** For atypical normal or abnormal actions, such as riding a bike, our method judges anomalies through test-time uniqueness analysis. Although surveillance videos often

cover a much longer temporal span and involve a larger number of people, there are some instances where only one person's activity is present. Moreover, when the person's movement trajectory is stable, we are also unable to infer through the self-inspection distance. In such special cases, it is not possible to get information from the activities of surrounding individuals, as illustrated in Fig. 6 (right). This issue will be explored in the future.

## A.5 MORE VISUALIZATION RESULTS

**We recommend readers watch our supplemental video for a more intuitive understanding of our method.** In addition, more visualization results are shown in Fig. 7 - Fig. 10. Each individual (blue skeleton) has a predicted anomaly score (red font), where the frame-level score (red line) is defined as the maximum among all individuals in that frame. The vertical axis represents the anomaly score, while the horizontal axis represents the frame index. Our approach demonstrates a robust ability to accurately identify anomalous events within novel environments, emphasizing the generalist nature of our model. In surveillance scenarios lacking training samples, our model supports a lightweight and effective approach to contribute the real-world video anomaly detection.

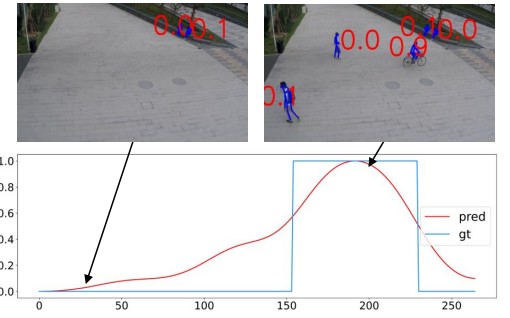

Figure 7: Visualization results on the ShanghaiTech dataset.

Figure 8: Visualization results on the UBnormal dataset.

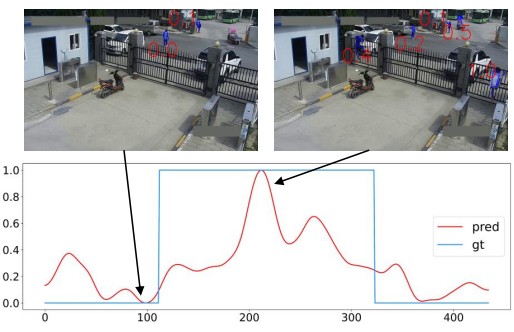

Figure 9: Visualization results on the NWPU dataset.

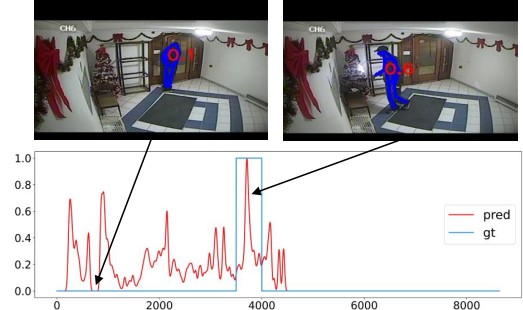

Figure 10: Visualization results on the UCF-Crime dataset.