# OpenReview forum: "Action Typicality and Uniqueness Learning for Zero-Shot Video Anomaly Detection"
_ICLR.cc/2025/Conference — ICLR 2025 Conference Withdrawn Submission_

### Official Review · Reviewer_GXqV · 2024-10-28

**Soundness:** 3
**Presentation:** 1
**Contribution:** 2
**Rating:** 5
**Confidence:** 3

**Summary:**

This paper proposes a new method for zero-shot video anomaly detection. Specifically, it first utilizes a module to align the skeleton and text for generalizable skeleton representations among normal and abnormal behaviors. Secondly, the high-confidence skeleton samples are selected to learn distributions. Then, the cross-person and self-inspection graphs are designed to calculate the uniqueness score for test-time adaption. Extensive experiments show the superior performance of the proposed method.

**Strengths:**

1. The experimental results achieve the sota.
2. The insights in test-time uniqueness analysis should be appreciated.
3. The experiments are comprehensive and solid.

**Weaknesses:**

1. The paper is not well-written. Some symbols and details are not described well.

**Questions:**

1. What does the $T$ in line 197 denote? The decompose process should be described in detail. What are the $g_i, g_j$ refer? And what are the $i$ and $j$ denote? They should be described clearly.
2. For typicality knowledge distillation, if I am not wrong, the equation seems to select one skeleton sequence with the most similar text label in each category, so how does this equation address the challenges proposed in Lines 225-227?
3. In line 281, how does the $A_i$ is associated with the person graph $p_i$ and timestamp $t_i$. The subscript can be denoted as the spatial index and temporal index concurrently. I am confused about that; it should be introduced clearly. The symbols in Eqa (8) and Eqa (9) are unclear; the subscripts for the person and timestep indexes are the same.
4. I am confused about the uniqueness scores. The uniqueness score is also high if the different normal actions exhibit low similarity/distance in different timesteps. So how do we ensure the high distance means anomaly accurately?
5. In the training phase, the dataset used is inconsistent with the previous methods. How does the author ensure the equability with other sota methods?

---

> ### Author Response · Authors · 2024-11-15
>
> Thank you for your insightful review. We have decided to **withdraw** for further improvement. Nevertheless, we will still provide some responses to address your concerns：
>
> Q1: 1) T is the length of a skeleton snippet; 2) $g_i$ is the text label of snippet I;
>
> Q2: Thank you for your suggestion and we will revise them. We intended to convey that we use skeleton-text similarity to filter for high-quality skeleton snippets.
>
> Q3: Thank you for your suggestion and we will revise them. During the preprocessing of skeleton snippets, $A_i$ is associated with a human trajectory index $p_i$ and timestamp $t_i$
>
> Q4: We use a top-k operation. Since normal snippets are common between individuals in real-world surveillance video, even if they have low similarity at different timesteps, top-k only picks snippets similar to themselves.
>
> Q5: We'll add experiments on this in the future version.

---

### Official Review · Reviewer_2Nd6 · 2024-10-30

**Soundness:** 2
**Presentation:** 2
**Contribution:** 2
**Rating:** 5
**Confidence:** 4

**Summary:**

This paper addresses zero-shot skeleton-based video anomaly detection. During training, the choice of a human behavior dataset is commendable, as it deviates from the commonly used datasets for Video Action Detection (VAD). The methodology of first learning visual features of skeletal points through an encoder and then extracting these features for input into a classifier (NF) is innovative. This approach avoids the previous practice of directly using skeletal point coordinates as classifier input to determine normality or abnormality.

**Strengths:**

The computation of two graphs, as seen in many papers, is well-executed here. One graph captures the motion relationships between adjacent individuals within the current data segment, while the other graph represents the motion state of the same individual across different data segments. This dual-graph approach provides a comprehensive understanding of human motion.

The model learns high-level semantic features, which enables a better representation of the data. This leads to improved performance in various tasks.

By utilizing an action recognition dataset, the model can learn a wider range of behavior categories, allowing for better generalization across multiple scenarios.

**Weaknesses:**

This paper lacks innovation. Language-guide action learning is a common technique in zero-shot action recognition, such as [1,2], and the author merely applied it to anomaly detection. The method presented is primarily based on STG-NF [3], which can be regarded as an extension of STG-NF incorporating language-guided and zero-shot capabilities.

The backbone of the method is the multi-scale CTR-GCN, while the backbone of STG-NF is ST-GCN. Therefore, it is unfair to compare the experimental results of this paper directly with those of STG-NF. I would like to know how the author's method performs on the popular setting of VAD.

The model struggles with anomaly detection when the motion trajectory becomes stable. This highlights a potential weakness in the model's ability to handle certain types of data patterns.

The visualization results in Figure 4 and Figure 6 are ugly. The font size used to represent the anomaly score is too large.

[1] Generative Action Description Prompts for Skeleton-based Action Recognition. ICCV, 2023.

[2] Zero-shot Skeleton-based Action Recognition via Mutual Information Estimation and Maximization. ACM MM, 2023.

[3] Normalizing Flows for Human Pose Anomaly Detection. ICCV, 2023.

**Questions:**

N/A

---

> ### Author Response · Authors · 2024-11-15
>
> Thank you for your insightful review. We have decided to **withdraw** for further improvement. Nevertheless, we will still provide some responses to address your concerns：
>
> Q1: Our work aims to extend the popular skeleton-only video anomaly detection framework to zero-shot tasks, addressing the limitations of current methods in low-level representation and domain-limited normality decision boundary.  Language-guide action learning is just one part of our framework.
>
> Q2: Table 3(a) shows the results of an experiment where we removed the aligned skeleton feature inputs and trained an STG-NF network using only our typicality data. Further details and experiments will be provided in a future version.
>
> Q3: Single-person stable motion scenarios are actually few in real-world uncropped surveillance video.
>
> Q4: Thanks for the suggestions. We'll polish them up.

---

### Official Review · Reviewer_kEwn · 2024-11-02

**Soundness:** 2
**Presentation:** 2
**Contribution:** 2
**Rating:** 3
**Confidence:** 4

**Summary:**

This paper successfully implements zero-shot Video Anomaly Detection (VAD) by incorporating scene-related knowledge for the anomalous detection of human actions in videos, utilizing a Kinect-based large action classification dataset to train the skeleton encoder. It proposes a skeleton-based zero-shot video anomaly detection framework that can adaptively determine anomaly boundaries based on the scene.

**Strengths:**

1. The paper introduces the **Language-Guided Typicality Modeling module**, which maps skeleton snippets to a semantic space and computes KL divergence loss with features derived from a text encoder. Notably, during the alignment of skeletons and text, the authors innovatively incorporate a Large Language Model (LLM) for label classification (normal/abnormal), akin to how humans consider contextual factors when judging typical behavior distributions.
2. The analysis of skeleton snippets addresses temporal and spatial inconsistencies for cross-scene recognition. During the inference phase, the method filters results based on the proximity of spatiotemporal context graphs.
3. The authors achieve state-of-the-art (SOTA) performance on the ShanghaiTech, UBnormal, NWPU, and UCF-Crime datasets without using training data.

**Weaknesses:**

1. I have concerns regarding the Knowledge Collection section. Is it feasible to accurately classify action class texts using only a single prompt? What if the LLM considers a normal behavior in surveillance video, such as riding a bicycle, but it is classified as abnormal in the actual test set?
2. As noted by the authors in the limitations section of the appendix, the skeleton-based methods used in this study have limitations related to detectors and trackers, and they struggle to effectively handle single-person stable motion scenarios.
3. The **Uniqueness Analysis** in the inference phase seems applicable to any skeleton-based method, which raises questions about its originality.

**Questions:**

In summary, I find this paper has notable merits, including the innovative introduction of LLM for scene-related anomaly detection and the fine-tuning of the encoder. However, I remain skeptical about whether this straightforward approach can effectively address cross-scene challenges and generalize to more scenarios in a zero-shot manner. Additionally, I believe the **Uniqueness Scores** could offer insights applicable to any skeleton-based method.

---

> ### Author Response · Authors · 2024-11-15
>
> Thank you for your insightful review. We have decided to **withdraw** for further improvement. Nevertheless, we will still provide some responses to address your concerns：
>
> Q1：1) The Kinect-400-skeleton dataset contains action labels for the skeletons. The LLM only needs to identify which actions are typical (normal) and atypical (anomalous); this is a relatively simple task for an LLM. 2) If our typicality scores exhibit bias, our uniqueness scores will compensate for deriving decision boundaries adapted to the target scenario.
>
> Q2: 1) Current skeleton extraction and tracking techniques are well-developed and continuously improving, leaving relatively few challenging cases. 2) single-person stable motion scenarios are also few in real-world uncropped surveillance video.
>
> Q3: Our method is the first application of uniqueness to skeleton-based video anomaly detection. Because anomalies are rare and unique, there appears to be little motivation to apply uniqueness in other tasks.

---

### Official Review · Reviewer_6adS · 2024-11-05

**Soundness:** 3
**Presentation:** 3
**Contribution:** 3
**Rating:** 5
**Confidence:** 2

**Summary:**

This paper presents a novel Zero-Shot Video Anomaly Detection (ZS-VAD) approach that leverages action typicality and uniqueness learning. The proposed model incorporates two main components. First, a Language-Guided Typicality Modeling module projects skeleton snippets into a semantic space, learning typical behavior distributions (normal and abnormal) informed by experiential language model knowledge. Second, a Test-Time Uniqueness Analysis module examines spatio-temporal variations among skeleton snippets, establishing adaptive anomaly boundaries for diverse scenes.

**Strengths:**

This work is a good attempt at leveraging both typicality and uniqueness, offering a solution that utilizes large language models (LLMs) alongside test-time adaptability.

**Weaknesses:**

1. In reality, anomalies in many scenarios are distinguished by their deviation from the typical data for that specific scene. Without any normal data, it becomes challenging to define scene-dependent anomalies, reducing the task to change detection based on shifts in motion patterns and prior knowledge (or common sense) of a set of pre-defined actions.
2. Besides, recognizing anomalies using only skeleton data loses the interaction information with the environment, which can hinder applications in real-world surveillance settings.
3. Moreover, this method depends heavily on skeleton estimation and skeleton-based action recognition methods. If we already have a well-defined action recognition method and powerful LLM, identifying anomalies or normal behavior types could be achieved directly. Without these prerequisites, however, the necessity of using this approach for anomaly detection becomes less clear. A deeper explanation is needed to clarify the rationale behind the skeleton-text alignment, the selection of typicality knowledge, and the process for learning typicality distributions.

**Questions:**

Please refer to the Weaknesses. And the number of parameters, training and inference costs should be given.

---

> ### Author Response · Authors · 2024-11-15
>
> Thank you for your insightful review. We have decided to **withdraw** for further improvement. Nevertheless, we will still provide some responses to address your concerns：
>
> Q1&Q2：Our work aims to extend the popular skeleton-only video anomaly detection framework to zero-shot tasks, addressing the limitations of current methods in low-level representation and domain-limited normality decision boundary. The complex relationships between behaviors and unseen, uncertain scenes will be investigated in future work.
>
> Q3: Our method uses LLMs only during training to acquire typicality knowledge for normalizing flow training; this can be viewed as LLM knowledge distillation. Consequently, our training pipeline's advantage is that we don't have to use LLM at inference time, ensuring a lightweight model suitable for widespread deployment.

---

### Note · Authors · 2024-11-15

I have read and agree with the venue's withdrawal policy on behalf of myself and my co-authors.